# A Genome-Wide Association Study on Liver Stiffness Changes during Hepatitis C Virus Infection Cure

**DOI:** 10.3390/diagnostics11081501

**Published:** 2021-08-20

**Authors:** Anaïs Corma-Gómez, Juan Macías, Antonio Rivero, Antonio Rivero-Juarez, Ignacio de los Santos, Sergio Reus-Bañuls, Luis Morano, Dolores Merino, Rosario Palacios, Carlos Galera, Marta Fernández-Fuertes, Alejandro González-Serna, Itziar de Rojas, Agustín Ruiz, María E. Sáez, Luis M. Real, Juan A. Pineda

**Affiliations:** 1Unidad Clínica de Enfermedades Infecciosas y Microbiología, Hospital de Valme, 41014 Sevilla, Spain; anais.corgo@gmail.com (A.C.-G.); juan.macias.sanchez@gmail.com (J.M.); martaffuertes@gmail.com (M.F.-F.); alextantalo@gmail.com (A.G.-S.); japineda@telefonica.net (J.A.P.); 2Departamento de Medicina, Universidad de Sevilla, 41009 Sevilla, Spain; 3Unidad de Enfermedades Infecciosas, Hospital Universitario Reina Sofía, Instituto Maimonides de Investigación Biomédica de Córdoba (IMIBIC), Universidad de Córdoba, 14004 Córdoba, Spain; ariveror@gmail.com (A.R.); arjvet@gmail.com (A.R.-J.); 4Unidad de Medicina Interna y Enfermedades Infecciosas, Hospital de La Princesa, 28006 Madrid, Spain; isantosg@hotmail.com; 5Unidad de Enfermedades Infecciosas, Hospital General Universitario de Alicante, 03010 Alicante, Spain; reus_ser@gva.es; 6Unidad de Patología Infecciosa, Hospital Universitario Alvaro Cunqueiro, Instituto de Investigación Sanitaria Galicia Sur (IISGS), 36312 Vigo, Spain; luis.morano.amado@gmail.com; 7Unidad de Enfermedades Infecciosas, Hospital Universitario Juan Ramón Jiménez, 21005 Huelva, Spain; merinolola080@gmail.com; 8Unidad de Enfermedades Infecciosas y Microbiología, Hospital Virgen de la Victoria, 29010 Málaga, Spain; rosariopalaci@gmail.com; 9Unidad de Medicina Interna, Hospital Clínico Universitario Virgen de la Arrixaca, 30120 Murcia, Spain; carlosgalerap@gmail.com; 10Fundació ACE—Institut Català de Neurociències Aplicades, Universitat Internacional de Catalunya (UIC), 08028 Barcelona, Spain; iderojas@fundacioace.org (I.d.R.); aruiz@fundacioace.com (A.R.); 11CIBERNED Centro de Investigación Biomédica en Red de Enfermedades Neurodegenerativas, Instituto de Salud Carlos III, 28029 Madrid, Spain; 12Centro Andaluz de Estudios Bioinformáticos (CAEBI, SL), 41013 Sevilla, Spain; mesaez@caebi.es; 13Departamento de Especialidades Quirúrgicas, Bioquímica e Inmunología, Universidad de Málaga, 29010 Málaga, Spain

**Keywords:** GWAS, hepatitis C virus infection, polymorphisms, sustained virological response, direct-acting antivirals, liver stiffness

## Abstract

Liver stiffness (LS) at sustained virological response (SVR) after direct-acting antivirals (DAA)-based therapy is a predictor of liver events in hepatitis C virus (HCV)-infected patients. The study aim was to identify genetic factors associated with LS changes from the moment of starting anti-HCV therapy to SVR. This prospective study included HCV-infected patients from the GEHEP-011 cohort who achieved SVR with DAA-based therapy, with LS pre-treatment ≥ 9.5 kPa and LS measurement available at SVR. Plink and Magma software were used to carry out genome-wide single-nucleotide polymorphism (SNP)-based and gene-based association analyses, respectively. The ShinyGO application was used for exploring enrichment in Gene Ontology (GO) categories for biological processes. Overall, 242 patients were included. Median (quartile 1, quartile 3) LS values at pre-treatment and at SVR were 16.8 (12, 28) kPa and 12.0 (8.5, 19.3) kPa, respectively. Thirty-five SNPs and three genes reached suggestive association with LS changes from the moment of starting anti-HCV therapy to SVR. GO categories related to DNA packaging complex, DNA conformation change, chromosome organization and chromatin organization were significantly enriched. Our study reports possible genetic factors associated with LS changes during HCV-infection cure. In addition, our results suggest that processes related to DNA conformation are also involved in these changes.

## 1. Introduction

The achievement of sustained virological response (SVR) among hepatitis C virus (HCV)-infected individuals is associated with a reduction in the incidence of liver complications, including hepatocellular carcinoma (HCC), as well as in all-cause mortality [1,2,3,4]. However, the risk of developing liver events does not disappear after SVR, especially among pre-treatment cirrhotic patients [3,5,6]. For that reason, patients with cirrhosis must undergo life-long surveillance for liver complications [7], and some authors extend this recommendation to individuals with pre-treatment advanced fibrosis [8].

Liver stiffness (LS), measured by vibration-controlled transient elastography (VCTE), is a surrogate marker of liver disease and strongly correlates with the emergence of liver-related events among both HCV-mono-infected and HIV/HCV-coinfected subjects [9,10,11]. Interestingly, SVR achievement also leads to a reduction of LS in this subset. Besides, this improvement is more pronounced in patients with higher pre-treatment LS values [12,13]. Recently, it has been reported that the LS value at SVR is also a strong predictor of liver disease outcome in HCV-infected patients with pre-treatment advanced fibrosis, irrespective of HIV-coinfection [14,15].

LS reduction from the moment of therapy starting to SVR, as other complex conditions, could be partially determined by genetic factors. However, to our knowledge, no genetic association study has been conducted on this trait yet. The identification of these host factors could be helpful to understand the molecular basis of this complex trait. Moreover, it could be useful to identify individuals with lower risk of hepatic events’ emergence after SVR, for whom surveillance programs for liver complications would no longer be cost-effective. In addition, these factors could be potential therapeutic targets among patients with a higher risk of clinical events after SVR. Due to this, the objective of this study was to identify genetic factors, by means of the genome-wide association study (GWAS) and gene-based association approach, related to LS changes from the moment of starting anti-HCV therapy to SVR among HCV-infected patients with pre-treatment advanced liver fibrosis. 

## 2. Materials and Methods

### 2.1. Design and Study Population

In this multicenter prospective cohort study, HCV-infected patients from the GEHEP-011 cohort were included (clinicaltrials.gov ID: NCT04460157). In this cohort, HCV-infected individuals, HIV-coinfected or not, who had received direct-acting antivirals (DAA)-based therapy after October 2011 at units of infectious diseases of 18 hospitals throughout Spain, are enrolled. Enrolment criteria in the cohort were: (1) have achieved SVR with DAA-based therapy, (2) have LS pre-treatment ≥ 9.5 kPa and (3) have a LS measurement at the SVR time point. 

Patients were included in this study if: (1) they had an available frozen blood sample before December 2017, (2) they were Caucasian and (3) they were non-familiarly related. 

### 2.2. Endpoint and Other Definitions

The primary endpoint was the percentage of LS change from the date of starting the DAA-based regimen to the SVR time point. 

LS was assessed by VCTE using a FibroScan^®^ (Echosens, Paris, France), according to a standardized procedure [16], within the 30 days before HCV therapy initiation (baseline) and at the SVR time point. These examinations were undertaken by an experienced operator at each participating center.

SVR was defined as undetectable HCV RNA 12 weeks after the end of anti-HCV therapy.

### 2.3. Genome-Wide Genotyping and Quality Controls

DNA isolation, genotyping methods, genotyping and samples’ quality controls, principal component (PC) analyses and SNPs imputation were performed as previously described elsewhere [17]. Briefly, the Axiom 815K Biobank array (Thermo Fisher, Waltham, Massachusetts, USA) and GeneTitan Multi-Channel instrument (Thermo Fisher) were used for sample genotyping. Samples with a call rate lower that 97% were excluded. SNPs with a call rate <95% or with a minor allele frequency (MAF) below 0.01 were removed. In addition, those individuals with heterozygosity rates greater than 0.35, or those who were related to other individuals in the sample (Identity by state (IBS) > 0.1875), were excluded. All these analyses were carried out using Plink software (version 1.9) (https://www.cog-genomics.org/plink (accessed on 1 January 2021)). PC analysis was run together with other genotype data of other populations obtained from phase 3 of the 1000 Genomes Project (http://www.internationalgenome.org/ (accessed on 1 January 2021)). Only individuals of Caucasian origin (using a threshold of 6 standard deviations from mean Caucasian PC values) were kept for further analyses.

### 2.4. Genome-Wide Association Analysis

Plink software was used to perform the GWAS, under the additive model, adjusted by the four main principal components, age (continuous), sex, liver stiffness at baseline, HIV infection and treatment regimen containing interferon. Significant *p*-value was established at 5 × 10^−8^ [18], whereas a *p*-value < 10^−5^ was considered as suggestive of statistical significance. The qqman R package (version 0.1.8) (https://CRAN.R-project.org/package=qqman (accessed on 1 January 2021)) was used for graphical representation of the GWAS single-locus analysis results (Manhattan plot). Genetic variants were annotated using the Variant Effect Predictor tool (version 104) [19]. The genomic inflation factor (ƛ) was also determined by Plink. 

### 2.5. Gene-Based Association Study and Enrichment Analyses

The Magma software (version 1.08) was used for calculating gene-wise statistics. This software detects multi-marker effects taking into account the physical distance and linkage disequilibrium between SNPs [20]. These analyses used a 50 kb upstream and downstream window around each gene to capture potential regulatory variants of these genes. The p_SNPwise_mean value calculated by the software was used for gene-based association analyses. This *p*-value was corrected by the number of genes analyzed by the software. Finally, 2 × 10^−6^ was the *p*-value threshold established for this study, whereas a *p* < 10^−4^ was considered as suggestive of statistical significance.

The ShinyGO application (version 0.61) [21] (http://bioinformatics.sdstate.edu/go/ (accessed on 1 January 2021)) was used for exploring enrichment in Gene Ontology (GO) categories [22,23] for biological processes using the 150 top genes obtained from the gene-based association analyses. Multiple testing correction was applied using the Benjamini–Hochberg method implemented in the application. We considered significant those processes with false discovery rate (FDR) *p*-value < 0.05. Only categories with a minimum of ten overlapping genes were selected.

### 2.6. Additional Statistical Analyses

Continuous variables were expressed as median (quartile 1, quartile 3) and categorical variables as frequencies (percentage). Comparisons of categorical variables were carried out using the Pearson chi-square test or the Fisher test. Quantitative variables were compared by means of Student’s *t*-test (data normally distributed) or Mann–Whitney U test (data not normally distributed). The Wilcoxon test was used to compare LS values at the baseline time point and at SVR. All these calculations were carried out using the SPSS software 25.0 (IBM Corporation, Somers, NY, USA).

## 3. Results

### 3.1. Study Population 

Among the 1035 individuals that constitute the GEHEP-011 cohort, a total of 261 (25.2%) subjects had samples available for this study. Among them, 10 (3.8%) individuals did not reach the DNA quality criteria to be massively genotyped, 4 (1.5%) showed relatedness with other included individuals and 5 (1.9%) did not have a Caucasian origin. Consequently, 242 (92.7%) subjects constituted the study population (GWAS population). 

The main characteristics of these individuals are depicted in Table 1. 

Direct-acting antiviral regimens used in the GWAS population for achieving sustained viral response are depicted in Appendix A. 

### 3.2. Changes in Liver Stiffness from Baseline to SVR

The median (quartile 1, quartile 3) value of LS at the baseline time point was 17 (12, 28) kPa, whereas the value at SVR was 12 (8.5, 19.3) kPa (*p* < 0.001). Overall, the median of the percentage of LS reduction from baseline to SVR was 28.8% (11.1%, 46.8%). More specifically, 27 (11.1%) individuals showed a LS increment of 17.8% (8.5%, 22.3%) at SVR, whereas 212 (87.6%) individuals experienced a LS reduction of 33.7% (16.2%, 48.4%). Three (1.2%) patients did not show a LS change from baseline to SVR.

### 3.3. Genome-Wide Association Study

Initially, 592,389 SNPs passed the genotyping quality controls. After imputation, a total of 6,939,676 variants were available for subsequent analysis. The GWAS population did not reveal an admixture in the principal component analysis (Appendix A). In addition, no overall inflation of the test statistic was observed (ƛ = 1.00) (Appendix A), supporting that systematic confounding factors were unlikely. 

There was not any SNP associated with the percentage of LS change from baseline to SVR at the *p*-value threshold established for GWAS (Figure 1). Nevertheless, 35 SNPs reached suggestive association with this endpoint (Table 2).

### 3.4. Gene-Based Association and Enrichment Analyses

The Magma software was used to carry out a gene-based association analysis. A total of 18,127 genes were ranked. None of them reached the multiple testing corrected *p*-value. Nevertheless, 3 genes reached suggestive statistical significance: LYPLAL1, PTGR1 and SLC8A3 (Appendix A). 

We analyzed if the best 150 ranked genes obtained in the gene-based association study (Appendix A) were significantly aggregated in specific categories of Gene Ontology for biological processes. Processes related with DNA packaging complex, DNA conformation change, chromosome organization and chromatin organization reached the established FDR *p*-value threshold (Table 3; Appendix A). Nevertheless, the same genes (*HIST2H2BE*, *HIST2H4B*, *SMC2*, *HIST2H3A*, *H2AFY*, *PRM2*, *PRM1*, *PRM3*, *CDAN1*, *TNP2*) were present in all these related biological processes.

## 4. Discussion

In this work, we have reported possible genetic factors involved in the LS changes from the moment of starting anti-HCV therapy to SVR. In addition, our data suggest that processes related to DNA conformation are involved in these changes.

Although the statistically significant *p*-value threshold established for the GWAS was not reached by any SNP, some suggestive associations deserve attention. On one hand, and with respect to the top SNPs identified in our GWAS, the strongest signals were linked to the *NDUF2* gene. It has been recently reported that the expression of Nduf2, a component of the complex I of the mitochondrial respiratory chain, was upregulated in the hepatic stellate cells (HSCs) from a rat model of alcohol-induced fibrosis. In this model, Nduf2 was involved in the regulation of fibrosis factors [24]. Other top SNPs were linked to *MAN1A1* and *AHRR* locus. MAN1A1 is a mannosidase with an important role in the formation of mature glycoproteins, protein folding and in misfolded protein degradation in eukaryotes [25]. Overexpression of MAN1A1 in a transgenic zebrafish model promoted the development of steatosis, inflammation and hepatocellular carcinoma formation, and also induced the overexpression of MM9 [26], a metalloprotease that seems to be strongly involved in liver fibrosis resolution [27]. AHRR represses signal transduction promoted by the Aryl-hydrocarbon receptor (AhR) [28]. AhR, highly expressed in the liver, is a xenobiotic receptor that senses environmental toxicants and regulates xenobiotic metabolism [29]. Recently, a controversial but important role of this protein in liver fibrosis has been suggested. Specifically, knockout of AhR in HSCs causes spontaneous liver fibrosis; in contrast, a non-toxic AhR agonist exhibited in vivo anti-fibrotic activity in mice [30]. Therefore, a dysregulation on AHRR could interact with the role of AhR in a possible anti-fibrotic activity. On the other hand, and regarding the gene-based association study, we identified the *LYPLAL1* gene as the top gene suggestively associated with the percentage of LS variations from baseline to SVR. Interestingly, a variant linked to this gene was related to histologic lobular inflammation/fibrosis in a GWAS performed on non-alcoholic fatty liver disease [31]. Due to all of these facts, future validations of these results in independent studies are warranted. 

Among HCV-infected individuals with a more advanced liver disease, liver stiffness normalization, as a reflection of liver injury improvement after HCV-infection cure, might not be achieved. This non-return point in the course of liver disease would be more frequent among infected patients who carry genetic risk factors for fibrosis progression. Therefore, it could be hypothesized that genetic factors involved in liver disease progression during HCV active infection could also be associated with LS changes observed during the HCV treatment. Two GWAS were carried out among Caucasian individuals in the setting of HCV active infection [32,33], and several SNPs and their linked genes were related to this trait. However, none of them appeared in the top list of SNPs or genes associated with the endpoint analyzed herein, nor did they show associations at nominal *p*-value (<0.05) (data not shown), suggesting that mechanisms involved in liver disease progression during HCV active infection are different to those involved in liver disease regression at SVR.

Interestingly, some of those top genes associated with the endpoint were grouped in linked GO categories related with the conformation of the DNA and chromatin. Recently, it has been reported that HCV infection induces genome-wide epigenetic changes through histone modifications that lead to changes in active and repressed chromatin and, consequently, reprograming of host gene expression. Besides, these changes persisted after HCV eradication with DAA-based treatment [34,35]. In addition, the persistence of this HCV-induced expression signature was more common in patients with pre-treatment liver fibrosis and was related to the risk of HCC after HCV-infection cure [34,35]. In light of these findings, our enrichment analysis results suggest the existence of a genetic susceptibility for such epigenetic modifications induced by the HCV infection. 

In spite of LS at SVR being a strong predictor of liver disease outcome in HCV-infected patients with pre-treatment advanced fibrosis [14,15], other factors have been independently associated with liver events’ occurrence. For instance, HIV-infection and non-genotype 3-HCV-infection have been related to a lower risk of HCC [36], whereas the anti-HCV therapy used as well as the presence of diabetes have been associated with incremented risk of HCC [37]. Therefore, it will be necessary to explore if the genetic factors described herein have a role in the occurrence of these events as well as to determine their real effect, taking into account all those non-genetic factors. 

This work has some limitations. Firstly, because the collection of samples was not mandatory to enter patients in the GEHEP-011 cohort, a relatively low number of individuals were included in our study. This fact, together with the possible polygenetic nature of the LS changes from baseline to SVR, would explain why no SNP reached the *p*-value threshold established for GWAS. However, the biological plausibility of some of the genes linked to those SNPs reported as suggestively associated with the main endpoint and, the concordance of the enrichment analysis results with the molecular mechanisms proposed for the persistence of the risk for liver events’ development in HCV-cured individuals, are strengths of this work. Second, unfortunately, we could not include a replication sample. Therefore, replication of the findings in other cohorts is needed to confirm these findings. Additionally, if our results are replicated, it would be necessary to identify whether the effect size of the genetic signals reported herein are homogeneous in the population sub-groups stratified according to HIV-infection or treatment received. It is important to note that we have not taken into account the duration of treatment and, therefore, we do not know if this issue could be affecting our results. However, all LS determinations at SVR were performed 12 weeks after the end of anti-HCV therapy. Moreover, it is known that the use of IFN could affect the regression of the LS after HCV-infection cure. Although a low proportion of patients were treated with DAAs combined with pegylated interferon, our results were corrected by this factor. Finally, the inclusion of HCV-infected individuals was conditioned by the availability of blood samples. Therefore, a selection bias cannot be excluded. In fact, our study population was younger than that of the entire cohort. Moreover, our study population had a higher proportion of HIV-infected subjects, and a higher proportion of subjects who received DAAs combined with pegylated interferon than that observed in the entire cohort. Nevertheless, our results were corrected by all of these factors. 

In conclusion, we have performed the first GWAS on LS values’ changes from baseline to SVR among HCV-infected patients, HIV-coinfected or not, with pre-treatment advanced liver fibrosis. Our results suggest that these changes in LS values could be partially conditioned by multiple gene variants. Although further studies will be necessary to confirm our results, our work provides clues about the possible molecular pathways involved in this condition.

## Figures and Tables

**Figure 1 diagnostics-11-01501-f001:**
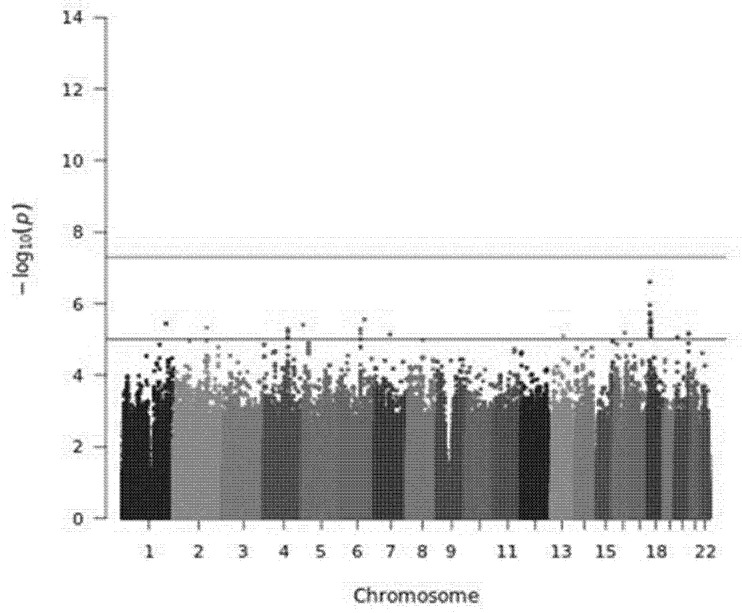
Manhattan plot of the GWAS on the percentage of LS changes at SVR. Horizontal lines correspond to of 1 × 10^−5^ and 5 × 10^−8^ *p*-values, respectively.

**Table 1 diagnostics-11-01501-t001:** Main characteristics of both the study populations and the entire GEHEP-011 cohort.

Variables	GWAS Population (*n* = 242)	Gehep-011 Entire Cohort (*n* = 1035)	*p*
Age, years ^†^	50 (46–53)	51 (47–55)	0.019
Males, *n* (%)	202 (83.5)	844 (81.5)	0.484
HIV infection, *n* (%)	174 (71.9)	667 (64.4)	0.028
PWID, *n* (%)	181 (74.8)	697 (67.3)	0.025
HCV genotype 3, *n* (%)	41 (16.9)	178 (17.2)	0.919
HCV viral load ^† ‡^	163 (68–398)	177 (59–415)	0.854
IFN-free treatment, *n* (%)	208 (86)	973 (94)	<0.001
Baseline liver stiffness (KPa) ^†^	116.8 (11.8–27.7)	16.8 (11.8–26.7)	0.487

PWID, People who injected drugs; HCV, Hepatitis C virus; IFN, interferon; kPa, Kilo Pascals. ^†^ Median (quartile 1–quartile 3). ^‡^ Expressed as 100,000 UI/mL.

**Table 2 diagnostics-11-01501-t002:** Best single-locus genetic association analysis results (*p* < 10^−5^) obtained by plink.

CHR	SNP	BP	A1	MAF	BETA (95%CI)	*p*	Gene
18	rs12606769	9012996	C	0.207	−13.27 (−18.16–−8.386)	2.48 × 10^−7^	*NDUFV2* ^†^
18	rs9959475	9009311	A	0.234	−12.14 (−16.89–−7.387)	1.10 × 10^−6^	*NDUFV2* ^†^
18	rs1553736	9012842	C	0.215	−12.21 (−17.09–−7.333)	1.80 × 10^−6^	*NDUFV2* ^†^
18	rs56232039	9013366	A	0.212	−12.23 (−17.14–−7.315)	2.05 × 10^−6^	*NDUFV2* ^†^
6	rs2649545	119771886	A	0.447	−10.98 (−15.45–−6.508)	2.77 × 10^−6^	*MAN1A1* ^†^
18	rs67749125	11936826	T	0.048	−25.16 (−35.44–−14.89)	2.91 × 10^−6^	*MPPE1* ^†^
18	rs8095587	11936664	A	0.048	−25.16 (−35.44–−14.89)	2.91 × 10^−6^	*MPPE1* ^†^
18	rs12326768	11925795	G	0.052	−24.06 (−33.96–−14.17)	3.37 × 10^−6^	*MPPE1* ^†^
18	rs72942777	9004175	T	0.268	−10.78 (−15.22–−6.344)	3.45 × 10^−6^	*NDUFV2* ^†^
1	rs188739258	208743339	T	0.013	−42.14 (−59.53–−24.75)	3.60 × 10^−6^	
5	rs62330020	311880	T	0.047	−22.65 (−32.03–−13.26)	3.94 × 10^−6^	*AHRR*
2	rs6437198	159613196	C	0.277	−11.38 (−16.13–−6.625)	4.70 × 10^−6^	*APL1* ^†^
18	rs12955366	11922625	C	0.056	−23.79 (−33.75–−13.83)	4.90 × 10^−6^	*MPPE1* ^†^
18	rs9951113	11927429	C	0.050	−24.12 (−34.22–−14.01)	4.95 × 10^−6^	*MPPE1* ^†^
6	rs9402699	99800122	A	0.277	−11.49 (−16.31–−6.662)	5.19 × 10^−6^	*FAXC* ^†^
4	rs8180156	115907555	C	0.110	16.62 (9.637–23.6)	5.22 × 10^−6^	*NDST4*
18	rs9962961	11936268	C	0.050	−24.11 (−34.24–−13.98)	5.25 × 10^−6^	*MPPE1* ^†^
18	rs112570549	11920486	T	0.051	−23.43 (−33.33–−13.53)	5.95 × 10^−6^	*MPPE1* ^†^
4	rs17623036	115902626	C	0.108	16.61 (9.57–23.66)	6.32 × 10^−6^	*NDST4* ^†^
16	rs74918996	53055101	C	0.022	−33.49 (−47.71–−19.27)	6.55 × 10^−6^	*CHD9* ^†^
6	rs6924993	99798835	G	0.281	−11.37 (−16.21–−6.538)	6.69 × 10^−6^	*COQ3* ^†^
20	rs6122460	62100105	A	0.096	−15.91 (−22.68–−9.141)	6.83 × 10^−6^	*EEF1A2* ^†^
18	rs11081454	9005806	A	0.257	−10.86 (−15.49–−6.238)	6.96 × 10^−6^	*NDUFV2* ^†^
20	rs310602	62109170	A	0.101	−15.78 (−22.51–−9.059)	7.07 × 10^−6^	*EEF1A2* ^†^
7	rs71537604	73067749	T	0.071	−18.85 (−26.9–−10.81)	7.30 × 10^−6^	*VPS37D* ^†^
18	rs56786794	11922278	A	0.057	−22.95 (−32.76–−13.14)	7.53 × 10^−6^	*MPPE1* ^†^
13	rs61971490	73064151	T	0.045	−24.11 (−34.45–−13.77)	8.10 × 10^−6^	
18	rs12960421	11923803	G	0.058	−22.89 (−32.72–−13.06)	8.23 × 10^−6^	*MPPE1* ^†^
20	rs4815993	7263756	A	0.348	10.09 (5.749–14.43)	8.62 × 10^−6^	
4	rs114558514	115943168	T	0.108	16.51 (9.389–23.63)	8.99 × 10^−6^	*NDST4* ^†^
4	rs74700222	115938513	G	0.108	16.51 (9.389–23.63)	8.99 × 10^−6^	*NDST4*
4	rs76144590	115936946	G	0.108	16.51 (9.389–23.63)	8.99 × 10^−6^	*NDST4*
4	rs77275720	115946034	C	0.108	16.54 (9.395–23.68)	9.26 × 10^−6^	*NDST4*
6	rs13201542	99934459	A	0.076	−17.47 (−25.04–−9.894)	9.96 × 10^−6^	*USP45*
6	rs28385588	99915661	G	0.076	−17.47 (−25.04–−9.894)	9.96 × 10^−6^	*USP45*

CHR, chromosome; SNP, single-nucleotide polymorphism; BP, base pair position according to UCSC genome browser (NCBI37/hg19) and dbSNP build 150; A1, minor allele; MAF, minor allele frequency; CI, confidence interval. ^†^ Closer gene within 100 kilobases.

**Table 3 diagnostics-11-01501-t003:** Enrichment analysis results obtained with the ShinyGO v0.61 application.

Process Description	GO Term	Number of Genes in the Term	FDR *p*-Value	Overloaded Genes
DNA packaging	GO:0006323	221	<0.001	*NCAPH*, *HIST2H2BE*, *HIST2H4B*, *SMC2*, *HIST2H3A*, *H2AFY*, *PRM2*, *PRM1*, *PRM3*, *CDAN1*, *TNP2*
DNA conformation change	GO:0071103	325	0.001	*NCAPH*, *HIST2H2BE*, *HIST2H4B*, *SMC2*, *HIST2H3A*, *H2AFY*, *PRM2*, *PRM1*, *PRM3*, *CDAN1*, *TNP2*, *RAD23B*
Chromosome organization	GO:0051276	1293	0.007	*H2AFY*, *STAG1*, *NCAPH*, *MSL2*, *HIST2H2AC*, *HIST2H2AB*, *HIST2H2BE*, *HIST2H4B*, *HIST2H2AA4*, *RAD23B*, *SMC2*, *RNF20*, *HIST2H3A*, *ATXN3*, *RBL2*, *PRM2*, *SETD5*, *PPM1D*, *PRM1*, *PRM3*, *CDAN1*, *TNP2*
Chromatin organization	GO:0006325	849	0.039	*H2AFY*, *MSL2*, *HIST2H2AC*, *HIST2H2AB*, *HIST2H2BE*, *HIST2H4B*, *HIST2H2AA4*, *RNF20*, *HIST2H3A*, *ATXN3*, *RBL2*, *SETD5*, *PPM1D*, *CDAN1*, *TNP2*

## Data Availability

The data presented in this study are available on request from the corresponding author and the Ethics Committee of the Hospital Universitario de Valme. The data are not publicly available due to the conditions established in the Spanish Law for Biomedical Research (Ley 14/2007, de 3 de julio).

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
