# Peer review of "A Genome-Wide Association Study on Liver Stiffness Changes during Hepatitis C Virus Infection Cure"

_diagnostics, 2021, doi:10.3390/diagnostics11081501_

Round 1

Reviewer 1 Report

General comments :

Line 36 : I am not quite sure what descriptive statistic was used to report LS at SVR: Median, Q1-Q3 (IQR?) are three distinct values (Median = Q2). Could the authors specify?

Line 83: Could the authors elaborate on the rationale for excluding patients of non-European ascendancy? Methods now exist to consider multiethnic genetic association and have shown to increase the power to detect low-frequency alleles of moderate effect that would suffer from low power in one population. 

Line 88-89: It would be useful for the reader to know what class of HCV-DAA was used to treat the patients included in the study. (Ie. Genotype 3 is harder to treat; IFN-based therapies have different regimens.)

Line 94-140: Methods used are sound and align with a widely accepted pipeline for GWAS and gene-association studies.

Line 141-192:

  • Did the author carry out any stratification of the results? Are the results the same in females, HIV-coinfected individuals, or IFN-treated individuals? As an aside, how long was the treatment for IFN-treated patients => could this be a cofounding factor since their treatment seems to lead to a lower decrease in liver stiffness. Please elaborate on the discussion: I believe that the paper would be enhanced if the authors would put the limitation of their study into perspective rather than putting together a list of limitations.
  • Did the author consider a two-loci or multiple-loci association test for SNP at suggestive threshold?
  • Did the author perform any kind of pathway enrichment/analysis or classification according to other gene ontology classifications? That would give the authors some perspective to discuss their findings in their biological context.

Overall, the authors put together a nice paper that is professionally presented and conform to the level expected for publication in MDPI. Although no statistically significant genes/SNP were identified, the paper is of interest to a specific type of audience and warrants publication following peer-review. 

Reviewer 2 Report

the authors report the first GWAS on LS values changes from base260 line to SVR among HCV-infected patients, HIV-coinfected or not, with pre-treatment ad261 vanced liver fibrosis. However findigns are not clear significant.

I nmy opinion to improve paper they shoudl perform a different approach analysis. Particulalry they should evaluate how many patietns had early or late HCC end how was LS in this patients and which gene was expressed. They also should remove HIV positivie patients since HIV infection determine activation in Fybroblast and therefore possible confounding effects on LS results.

Furhter should be improved introduction and discussion focusing more attention on the real findings

Moreover details about different DAAs regimen should be improved

Round 2

Reviewer 2 Report

The paper in the present form has been improved however it coudl better clarify some issues.

1) In the introduction as well as troughout the paper the author's should considere previous research focusing on factors related to liver stifness and hcc after SVR (

Rinaldi L, Nevola R, Franci G, Perrella A, Corvino G, Marrone A, Berretta M, Morone MV, Galdiero M, Giordano M, Adinolfi LE, Sasso FC. Risk of Hepatocellular Carcinoma after HCV Clearance by Direct-Acting Antivirals Treatment Predictive Factors and Role of Epigenetics. Cancers (Basel). 2020 May 26;12(6):1351. doi: 10.3390/cancers12061351. PMID: 32466400; PMCID: PMC7352473.) or Rinaldi L, Perrella A, Guarino M, De Luca M, Piai G, Coppola N, Pafundi PC, Ciardiello F, Fasano M, Martinelli E, Valente G, Nevola R, Monari C, Miglioresi L, Guerrera B, Berretta M, Sasso FC, Morisco F, Izzi A, Adinolfi LE. Incidence and risk factors of early HCC occurrence in HCV patients treated with direct acting antivirals: a prospective multicentre study. J Transl Med. 2019 Aug 28;17(1):292. )      
